# Immediate Effects of Myofascial Release on the Thoracolumbar Fascia and Osteopathic Treatment for Acute Low Back Pain on Spine Shape Parameters: A Randomized, Placebo-Controlled Trial

**DOI:** 10.3390/life11080845

**Published:** 2021-08-18

**Authors:** Andreas Brandl, Christoph Egner, Robert Schleip

**Affiliations:** 1DIPLOMA Hochschule, 37242 Bad Sooden-Allendorf, Germany; brandl.andreas@stud.diploma.de (A.B.); christoph.egner@diploma.de (C.E.); 2Conservative and Rehabilitative Orthopedics, Department of Sport and Health Sciences, Technical University of Munich, 80333 Munich, Germany

**Keywords:** thoracolumbar fascia, myofascial release, osteopathy, leg length discrepancy, kyphotic angle, lordotic angle

## Abstract

Background: Spine shape parameters, such as leg length and kyphotic or lordotic angle, are influenced by low back pain. There is also evidence that the thoracolumbar fascia plays a role in such pathologies. This study examined the immediate effects of a myofascial release (MFR) technique on the thoracolumbar fascia and of an osteopathic treatment (OMT) on postural parameters in patients with acute low back pain (aLBP). Methods: This study was a single-blind randomized placebo-controlled trial. Seventy-one subjects (43.8 ± 10.5 years) suffering from aLBP were randomly and blindedly assigned to three groups to be treated with MFR, OMT, or a placebo intervention. Spinal shape parameters (functional leg length discrepancy (fLLD), kyphotic angle, and lordotic angle) were measured before and after the intervention using video raster stereography. Results: Within the MFR group, fLLD reduced by 5.2 mm, *p* < 0.001 and kyphotic angle by 8.2 degrees, *p* < 0.001. Within the OMT group, fLLD reduced by 4.5 mm, *p* < 0.001, and kyphotic angle by 8.4°, *p* = 0.007. Conclusion: MFR and OMT have an influence on fLLD and the kyphotic angle in aLBP patients. The interventions could have a regulating effect on the impaired neuromotor control of the lumbar muscles.

## 1. Introduction

There is ongoing interest in fascia research, particularly as a possible cause of low back pain (LBP) [1]. The pain is likely triggered by nociceptors in the thoracolumbar fascia (TLF) [2]. LBP is one of the main reasons for visiting manual therapists, orthopedists, and osteopaths [3]. Acute LBP (aLBP) is etiologically differentiated from chronic LBP. The current medical assumption is that the cure rate of aLBP is 90% within six weeks and only 2–7% become chronic [3]. However, this time period is based on measuring the time between the doctor’s visit and return to work and does not describe the actual duration of pain [4]. In a meta-analysis of 11 studies (3118 patients), Itz et al. [5] documented that spontaneous recovery of aLBP patients in the first three months was only 33%. Nevertheless, 65% still suffer from LBP one year after the onset of pain. Pengel et al. put the recurrence rate with renewed disability at 33% of the aLBP cases [6], making it a critical cost and resource factor in health care systems in industrialized countries [7].

Functional leg length discrepancies (fLLD) are suspected to be involved in the development of LBP [8,9,10,11]. LBP patients show a greater tendency for asymmetry in motor control in the form of pathological muscle activation than healthy individuals [12]. The TLF, as a central biomechanical part of the pelvic corset system, influences pelvic statics. It can be concluded that other components, including postural parameters such as fLLD or altered kyphosis or lordosis angles, are also involved in these mechanisms [10,13] and may influence each other through myofascial chains [14]. The fasciae represent the connection between skin, muscles, and bones in a kind of tensegrity structure [1]. Possible treatment approaches for these structures may include myofascial release (MFR) techniques as a single intervention or individual osteopathic manipulative treatment (OMT). Improvements in pain and disability have been reported with the use of MFR as a treatment for LBP [15,16,17]. Changes in spinal shape parameters and TLF biomechanical behavior have also been reported [16,18,19]. Numerous studies similarly document significant reductions in pain and disability through OMT [17,20], as well as changes in spinal shape parameters and functional status [17,20,21].

However, the mechanisms behind the changes in postural parameters caused by manual techniques in LBP are poorly understood [10,22]. Moreover, even small changes in spinal shape could lead to long-term pathologies [23]. Therefore, the aim of this study was to investigate the immediate effects of MFR on the TLF and of OMT on spinal shape parameters with a focus on biomechanical changes. In a previous pilot study testing the feasibility of this study, MFR treatment showed promising changes in the myofascial chain system [24].

## 2. Materials and Methods

### 2.1. Study Design Overview

The study is a single-blind placebo-controlled randomized trial with three groups. Measurements were taken pre- and post-intervention conforming to the SPIRIT guidelines [25]. The study protocol was retrospectively registered with the German Clinical Trials Register (DRKS00024122) on 22.02.2021. The study has been reviewed and approved by the ethical committee of the Osteopathic Research Institute in Hamburg, Germany (No. 019-11), has been carried out in accordance with the declaration of Helsinki, and has obtained informed consent from the participants [26].

### 2.2. Setting and Participants

This study was conducted in an osteopathic practice in a medium-sized city in southern Germany. In a previous pilot study, power and sample size calculations were carried out and the number of participants was set at 25 per group [24]. The acquisition was carried out via direct contact, a notice board, and the distribution of information material in the practice or to acquaintances. The study design envisaged carrying out the study in running practice, which is why the participants were continuously recruited during the study period. All test persons received a voucher for a preventive service of their own choice in the amount of 30 euros.

#### 2.2.1. Inclusion Criteria

Inclusion criteria were: (a) acute lumbar back pain (aLBP) as defined by the European guidelines for the management of acute low back pain [27]; (b) a minimum score of 10 on the Oswestry disability questionnaire in the German version (ODQ-D) [28]; (c) a minimum score of 3 on the visual analogue scale (VAS) [29]; (d) less than 6 weeks pain duration; (e) female or male subjects aged 18 to 60 years; (f) prone position for 15 min must be pain-free for the subjects.

#### 2.2.2. Exclusion Criteria

Exclusion criteria were (a) generally valid contraindications to physiotherapeutic and osteopathic treatments of the lumbar spine and pelvis; (b) rheumatic diseases; (c) taking medication that affects blood coagulation; (d) taking muscle relaxants; (e) skin changes (e.g., neurodermatitis, psoriasis, urticaria, decubitus ulcers); (d) surgery or other scars in the lumbar region between Th12 and S1.

### 2.3. Randomization and Interventions

The volunteers were first screened for eligibility by the investigator. These were assigned covertly to the MFR, OMT, or placebo (PLC) group using block randomization. The randomization was carried out on the internet with the Application Research Randomizer, version 4.0 [30]. The subjects were not given any information by the investigator regarding their group membership or the intervention that was being delivered. The investigator then carried out the initial measurements. The subjects received the respective group-specific intervention from an osteopath who had more than 10 years professional experience in manual therapy and a master’s degree. This was followed by the post-intervention measurement. This was only initiated by the investigator. The actual recording of the parameters was done automatically, after a time delay, by a computer system. The examination and treatment were carried out by the owner of the individual practice for osteopathy (AB).

#### 2.3.1. Myofascial Release Intervention

The MFR group received an intervention as described by Chila and O’Connell [31]. Here, the subject is in a prone position with the arms at the sides of the body and the legs parallel to each other. The head is in a neutral position; the face lies in a recess in the head section of the therapy table. The patient is undressed to such an extent that the TLF between Th12 and S1 is accessible. The therapist stands contralateral to the side to be treated, at the level of the subject’s iliac crest (Figure 1). The therapist’s cranial hand is positioned thenar-sided immediately adjacent to the lumbar spine with extensive contact at the TLF at the level of L1 to L4 and acts as a palpation hand. The caudal hand doubles the palpation hand and initiates a direct stretch of the fascia laterally to a noticeable tissue resistance. The therapist follows the creep of the myofascial tissue to initiate further stretching of the TLF [1]. The applied force on the tissue is only moderate, ranging from 25 to 35 N and acting tangentially laterally in the direction of the abdominal muscles. The usual force applied during an MFR treatment was previously trained by the therapist using a phantom pad placed over a highly sensitive force plate. The training was considered sufficient when the blinded practitioner was able to perform 30 stroke applications on the force plate, all of which were within this force range and had no outliers. This training was repeated once per day during data collection. The duration of the entire technique is 60 to 90 s. However, the decisive factor for the effect is not so much the period of time over which the technique is practiced, but the occurrence of a myofascial release. Ajimsha et al. define this as the restoration of the optimal length of myofascial tissue structures, their functional improvement, and the reduction of pain in them [32]. The therapist had experience in this specific application of MFR with this pressure for more than 10 years.

#### 2.3.2. Osteopathic Manipulative Treatment

The subjects of the OMT group underwent an osteopathic examination. Manual treatment of the structures identified as dysfunctional was carried out, individually adapted to the symptomatology of the individual subjects, as corresponds to the general procedure in osteopathic practice [31]. For example, pelvic asymmetry was treated by finding restrictions in the myofascial chains [14], monosegmental blockages of a vertebra via the muscular connections and with thrust techniques [33], or restrictions in the galea aponeurotica by treating the area with adhesions. For example, pelvic asymmetry was treated by finding restrictions in the myofascial chains [14], monosegmental blockages of a vertebra via the muscular connections and with pushing techniques [33], or restrictions in the galea aponeurotica by treating the area with adhesions. This means that although the subjects had acute low back pain, the osteopathic principles were applied, i.e., not necessarily only the painful region was treated, as in the MFR group.

The interventions used included direct techniques such as high-velocity, low-amplitude, muscle energy, fascia, indirect techniques such as functional techniques or balanced ligamentous tension, as well as visceral and cranial techniques [31]. The applied forces ranged between 3 N (cranial techniques) and 50 N (muscle energy techniques) for each hand, except for high-velocity, low-amplitude techniques with short peak forces due to impulses up to 300 N [33]. Subjects were assessed for 15 min according to their individual symptoms and treated for 30 min according to the findings by a therapist who had more than 10 years of professional experience in this procedure and a master’s degree in osteopathy.

#### 2.3.3. Control

The subjects in the PLC group were in a prone position with their arms at their sides and their legs parallel to each other. The head was in a neutral position; the face was in a recess of the head part of the therapy table. The subject was undressed to such an extent that the TLF between Th12 and S1 was accessible. The therapist stood contralateral to the side to be treated, at the level of the subject’s iliac crest (Figure 1). The therapist’s cranial hand was thenar-sided immediately adjacent to the lumbar spine with extensive contact at the TLF at the level of L1 to L4 and acted as a palpation hand. The caudal hand duplicated the palpation hand. Instead of an MFR, both hands were placed on the tissue with minimal pressure, ranging between 4 and 6 N for both hands, and left there for 90 s.

### 2.4. Outcomes

By means of VRS measurement, the fLLD in mm and the maximum kyphotic and lordotic angle in degrees were determined. These parameters have been shown to be valid and reliable in a paper by Degenhardt et al. [34]. For the fLLD, the authors gave an ICC = 0.84; 95% confidence interval (CI) = 0.73–0.90 and for the smallest detectable change (SDC) 4.21 mm, for the kyphotic angle an ICC = 0.96; 90% CI 0.92–0.97 and a SDC of 3.19°, for the lordotic angle an ICC = 0.91; 90% CI 0.83–0.94 and a SDC of 4.24°.

### 2.5. Statistical Analyses

For all parameters, the standard deviation (SD), standard error of the mean (SEM), the mean, the 95% CI, and the minimum (min) and maximum values (max) were determined. The outcome variables were normally distributed as assessed by the Shapiro–Wilk test (*p* > 0.05). The homogeneity of the error variances between the groups was fulfilled for all these variables according to Levene’s test (*p* > 0.05). Between-subject, within-subject, and interaction effects were tested for significance using a mixed ANOVA. Post hoc analysis was conducted using Tukey’s HSD test. The α-level was adjusted using Bonferroni correction. The significance level was set at *p* = 0.05.

Libreoffice Calc version 6.4.7.2 (Mozilla Public License v2.0) was used for the descriptive statistics. The inferential statistics were carried out with the software R, version 3.4.1 (R Foundation for Statistical Computing, Vienna, Austria). Statistical power was calculated with GPower (© 1992–2014 Franz Faul, University of Kiel, available at https://www.psychologie.hhu.de/arbeitsgruppen/allgemeine-psychologie-und-arbeitspsychologie/gpower, accessed on 18 August 2021).

## 3. Results

The anthropometric data and baseline characteristics are shown in Table 1. Of 83 subjects screened between 29/07/2019 and 13/03/2020, 71 met eligibility criteria and received the interventions or sham treatment (Figure 2). No subject was unblinded accidentally or in any other way.

The mixed ANOVA showed significant interactions between the measurement time points and the study groups for the fLLD measurement (F(2, 68) = 9.67, *p* < 0.001, partial η^2^ = 0.22, 1-ß err prob > 0.99). After treatment, the groups differed significantly (*p* < 0.001). According to Tukey’s HSD test, both the MFR group (5.2 mm, *p* < 0.001) and the OMT group (4.5 mm, *p* = 0.004) were significantly different from the PLC group. There was no significant difference between the MFR and OMT groups (*p* = 0.23).

There was also a significant interaction for the kyphotic angle (F(2, 68) = 3.30, *p* = 0.04, partial η^2^ = 0.09, 1-ß err prob > 0.99). After treatment, the groups differed significantly (*p* = 0.005). The MFR group (8.23, *p* = 0.008) and the OMT group (8.42°, *p* = 0.006) were significantly different from the PLC group, as shown by Tukey’s HSD test. There was no significant difference between the MFR and OMT groups (*p* = 0.99).

There was no significant interaction for the lordotic angle (F(2, 68) = 1.87, *p* = 0.16, partial η^2^ = 0.045, 1-ß err prob = 0.90) and no significant main effect for time (F(1, 140) = 0.63, *p* = 0.43, partial η^2^ = 0.004, 1-ß err prob = 0.14) and group (F(1, 139) = 0.4, *p* = 0.67, partial η^2^ = 0.006, 1-ß err prob = 0.19).

Within the MFR group, there was a statistical effect of intervention on fLLD (F(1, 46) = 27.3, *p* < 0.001, partial η^2^ = 0.37, 1-ß err prob > 0.99) and kyphotic angle (F(1, 46) = 12.76, *p* < 0.001, partial η^2^ = 0.22, 1-ß err prob > 0.99). Within the OMT group, there was a statistical effect of the intervention on fLLD (F(1, 46) = 12.87, *p* < 0.001, partial η^2^ = 0.22, 1-ß err prob > 0.99) and kyphotic angle (F(1, 46) = 12.89, *p* < 0.001, partial η^2^ = 0.22, 1-ß err prob > 0.99). Within the PLC group, there was no statistical effect of the sham intervention on fLLD (F(1, 44) = 0.074, *p* = 0.787, partial η^2^ = 0.002, 1-ß err prob = 0.09) and kyphotic angle (F(1, 44) = 0.101, *p* = 0.75, partial η^2^ = 0.002, 1-ß err prob = 0.09).

The SDC for fLLD in the MFR group was exceeded in 19 out of 24 cases (79.2%) and for the kyphotic angle in 21 out of 24 cases (87.5%). Within the OMT group, the SDC for the fLLD was exceeded in 16 out of 24 cases (66.6%) and for the kyphotic angle in 18 out of 24 cases (75%).

The changes between the baseline measurement and the measurement after treatment, or sham treatment, are shown in Table 2 and Figure 3.

## 4. Discussion

This study examined the immediate effects of MFR on the TLF and of individual OMT on spine shape parameters. The main results showed a clear difference between the intervention methods and the sham treatment. Here, in addition to the alteration of the skin receptors, mechanoreceptors in the fascial tissue under the skin could be stimulated (e.g., in the epi/peri/endo-mysium, fascia profunda, tendons, and joint capsules). In contrast, the much lighter touch of the PLC treatment is likely to act only on cutaneous receptors. Fascial mechanoreceptors could trigger changes in muscle tone, fluid hydration, as well as neurological effects, which was likely achieved through the interventions in this study [35,36].

The measured parameters, except for the lordotic angle, showed high modifiability by the interventions. The fLLD and the kyphotic angle could be significantly reduced. A post hoc power analysis determined a test strength of 1-β err prob > 0.99 for the interaction effects of these parameters, underscoring that the effects were not random, and the sample size was chosen large enough to achieve statistical power. A large effect size was obtained for both treatments (MFR: partial η^2^ = 0.37; OMT: partial η^2^ = 0.22). This is consistent with a previous study by Barnes et al. that examined the effect of a range of MFRs on fLLD (partial η^2^ = 0.47) [22]. For manual therapy, medium effects on fLLD (partial η^2^ = 0.11) are reported by Jeon et al. [37] and no effects for spinal manipulation by Schmidt et al. (partial η^2^ = 0.0025) [38]. However, there are few studies with small numbers of subjects that have investigated the effect of manual intervention on spinal shape parameters. It would be interesting to investigate the benefits of OMT and MFR obtained here compared to other techniques in future high-quality studies that also consider long-term changes.

Barnes et al. discussed that targeted MFR influences the myofascial structures of the pelvis so that the os ilium rotates toward symmetry relative to the os sacrum [22]. Among manual therapists, the hypothesis of leg length change due to anteroposterior ilium rotations is widespread [22,31,39]. However, due to the low sacroiliac joint mobility, which allows a maximum fLLD change of less than 1 mm after taking into account the biomechanical components [40], this concept seems too reduced to explain the results of this study, with fLLD changes of 5.2 mm within the MFR group and 4.5 mm within the OMT group. Furthermore, this does not take into account other spinal shape parameters, such as the kyphotic angle, which was significantly reduced (MFR group: 6.4°; OMT group: 8.4°). Stecco et al. emphasize the role of the connection of the fascia to the muscle spindles and its influence on motor control [41]. In the case of LBP, the muscle spindles could be blocked by reduced gliding and increased adhesion of the TLF (Figure 4).

According to the asymmetrically altered motor control and activation of the back muscles, especially the erector spinae muscle (ES), unilateral functional scoliosis (fSC) could cause fLLD. Sheha et al. found a high prevalence of fSC in fLLD in a systematic review [42]. The direct modifiability of fLLD by manual intervention, as observed in the study, makes it debatable whether fSC are less an effect than a cause of this phenomenon (Figure 5).

In most people, a right-dominant motor postural pattern is present, which leads to a certain physiological asymmetry. This asymmetry is more prevalent in LBP patients, inducing a unilateral pathological activation of the lumbar musculature [12]. Wilczyński et al. found significant correlations between increased ES neuromuscular activity and the Cobb angle of scoliotic postural change [43]. They conclude from their results that idiopathic scoliosis is frequently the consequence of an asymmetrically increased muscle tone of the ES. Based on this mechanism, the therapeutic interventions of both the MFR and OMT groups could have a direct influence on the neuromuscular activation of the ES in the present study. The results suggest that the investigation of neuromuscular aspects is particularly promising [16]. Further work in this regard is eagerly awaited.

The study presented here fulfilled the criteria of an RCT and had the strengths mentioned above, especially due to the generation of a sufficient number of cases but some limitations. No follow-up measurement was carried out in the present study. In this respect, the clinical relevance of the post-intervention results can only be assumed. The significant effects of the interventions therefore cannot provide any information about longer-term changes in spinal shape parameters, which should be the task of future work. Furthermore, no explicit distinction was made between subjects with specific and nonspecific LBP. Only 15% of all LBP patients have detectable pathologies that lead to the diagnosis of specific LBP [44]. This classification is due to the circumstance that in most cases of LBP, there are still no generally accepted diagnostic methods to assess the causal origin of the pain [4]. Even in patients with demonstrable pathology, the causal relationship between pathology and pain is unclear. Rheumatic diseases, the use of anticoagulants, muscle relaxants, skin changes, and scars were considered exclusion criteria for this study, which probably increased the percentage of subjects with nonspecific aLBP well above 85%, as at least all rheumatic or surgically treated patients with specific aLBP were not examined.

The results of this study should be seen in the light of the aim to investigate the immediate effects of two intervention methods on biomechanical changes in spinal shape parameters and to compare them with a sham intervention. We hypothesize that a possible mechanism behind the treatment effects could be higher pressure stimulating the fascial mechanoreceptors under the skin, rather than the much softer touch of the sham intervention.

## 5. Conclusions

Both MFR on the TLF and OMT showed immediate effects on the spine shape parameters fLLD and kyphotic angle. The fLLD and kyphotic angle reduced significantly within the MFR and OMT groups. These values also exceeded SDC in more than threequarters of the cases, demonstrating that these effects are also clinically relevant. Further research should seek to consider the neuromuscular aspects of these associations.

## Figures and Tables

**Figure 1 life-11-00845-f001:**
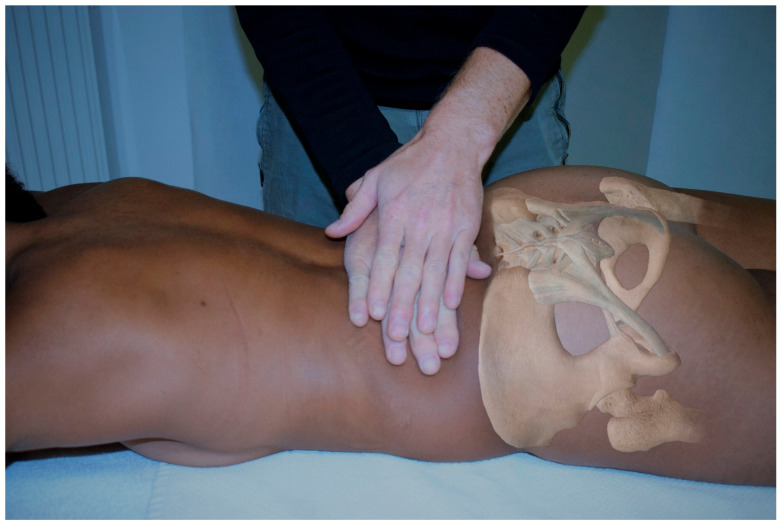
MFR and PLC treatment at the TLF. To illustrate the therapist’s hand placement, the participant placed her arms on the head of the therapy table.

**Figure 2 life-11-00845-f002:**
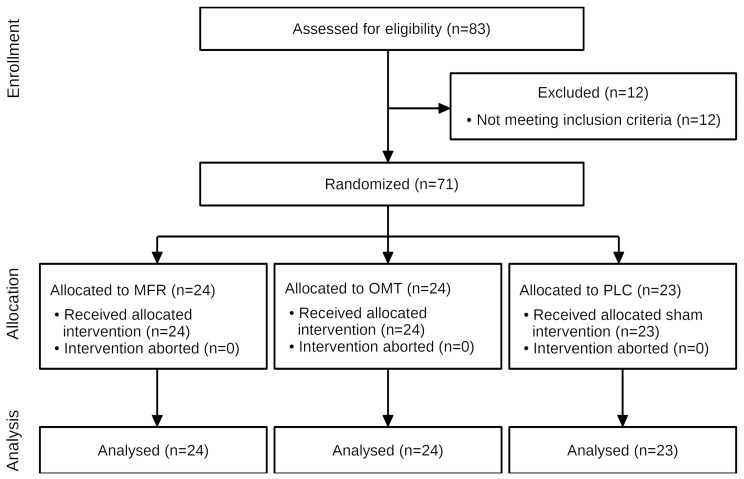
Flow diagram of the study. MFR, myofascial release; OMT, osteopathic manual treatment; PLC, placebo. “Loss to follow-up” has been omitted as it is not relevant to the study design.

**Figure 3 life-11-00845-f003:**
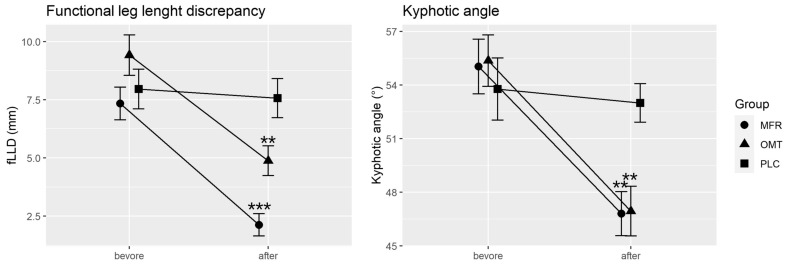
Change between baseline and after treatment. MFR, myofascial release; OMT, osteopathic manipulative treatment; PLC, placebo; fLLD, functional leg length discrepancy. Error bars represent the standard error of the mean. Significant at the level ** <0.01; *** <0.001.

**Figure 4 life-11-00845-f004:**
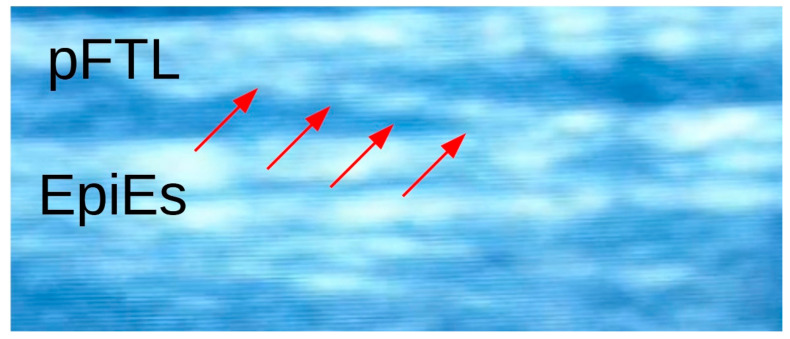
Ultrasound picture of an adhesion of the TLF. pFTL, posterior layer of the thoracolumbar fascia; EpiES, epimysium of the erector spinae muscle. The red arrows show a clear adhesion between the fascial layers.

**Figure 5 life-11-00845-f005:**
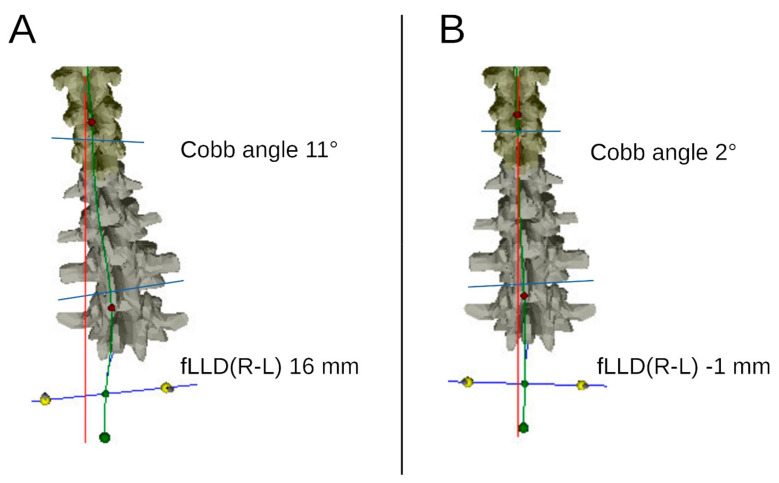
Video raster stereography of a patient with temporary functional scoliosis. fLLD(R-L) functional leg length discrepancy between the right and the left leg. The temporary functional scoliosis of 11° and the functional leg length discrepancy of 16 mm (**A**) were corrected almost completely after OMT treatment (**B**).

**Table 1 life-11-00845-t001:** Baseline characteristics.

Baseline Characteristics	MFR Group (*n* = 24) Mean ± SD	OMT Group (*n* = 24) Mean ± SD	PLC Group (*n* = 23) Mean ± SD	Total (*n* = 71) Mean ± SD
Gender (men/woman)	12/12	14/10	8/15	34/37
Age (years) min–max (years)	45.7 ± 9.4 20.4–59.9	43.2 ± 11.4 20.5–59.2	42.3 ± 10.6 23.9–59.3	43.8 ± 10.5 20.4–59.9
Height (m) min–max (m)	1.72 ± 0.1 1.60–1.92	1.75 ± 0.1 1.61–1.89	1.70 ± 0.1 1.59–1.89	1.72 ± 0.1 1.59–1.92
Weight (kg) min–max (kg)	75.8 ± 13.4 47–97	78.3 ± 14.3 60–110	72.3 ± 9.6 60–92	75.5 ± 12.7 47–110
BMI (kg/m^2^) min–max (kg/m^2^)	25.5 ± 3.6 17.9–35.2	25.5 ± 3.5 18.7–33.9	25.2 ± 3.7 21.7–35.2	25.4 ± 3.5 17.9–35.2
fLLD (mm) min–max (mm)	7.3 ± 4.0 0–13	9.4 ± 5.0 1–18	8.0 ± 4.9 0–17	8.2 ± 4.7 0–18
ODQ-D (0–100) min–max	23.2 ± 13.5 10–68	21.8 ± 11.0 11–48	22.6 ± 10.3 10–46	22.5 ± 11.5 10–68
VAS (0–10) min–max	5.0 ± 2.1 3–10	5.5 ± 1.9 3–10	4.7 ± 1.4 3–9	5.1 ± 1.8 3–10
Pain duration (days) min–max	11.7 ± 6.8 1–24	10.1 ± 8.5 1–29	14.4 ± 6.9 2–29	12.0 ± 7.6 1–29

SD, standard deviation; *n*, number; MFR, myofascial release; OMT, osteopathic manipulative treatment; PLC, placebo; fLLD, functional leg length discrepancy; ODQ-D, Oswestry disability questionnaire in the German version; VAS, visual analogue scale.

**Table 2 life-11-00845-t002:** Changes between baseline and after treatment.

Outcome	MFR Group (*n* = 24) Mean (95% CI)	OMT Group (*n* = 24) Mean (95% CI)	PLC Group (*n* = 23) Mean (95% CI)
fLLD (mm)	−5.2 (−8.8–−1.6) ***	−4.5 (−8.1–−1.0) **	−0.4 (−4.0–3.2)
Kyphotic angle (°)	−8.23 (−15–−1.4) **	−8.42 (−15–−1.6) **	−0.8 (−7.7–6.1)
Lordotic angle (°)	1.5 (−5.8–8.7)	−5.0 (−12.3–2.3)	0.1 (−7.4–7.5)

95% CI: 95% confidence interval. *n*, number; MFR, myofascial release; OMT, osteopathic manipulative treatment; PLC, placebo; fLLD, functional leg length discrepancy. Significant at the level ** <0.01; *** <0.001.

## Data Availability

Data can be made available by the author upon request.

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
