# Peer review of "Immediate Effects of Myofascial Release on the Thoracolumbar Fascia and Osteopathic Treatment for Acute Low Back Pain on Spine Shape Parameters: A Randomized, Placebo-Controlled Trial"

_life, 2021, doi:10.3390/life11080845_

Round 1

Reviewer 1 Report

The introduction sections requires a review and modifications . I appreciate the authors contribution and work in this manuscript, but the introduction requires grammatical modifications.

In addition, there is not a clear review of relevant literature on therapeutic benefits of the relevant manual therapy techniques used. This should be provided as justification to the study.

Methods are clearly outlined. However one potential issue is the control of force applied during the treatment. Whilst this is described, how was it controlled? What was the background to therapists performing the technique which might influencing reliability

Effect sizes should be included and interpreted ie establishing clinical worthwhile benefits of treatment relative to effects of other therapies

The authors provide a series of experimental limitations including lack of follow up. The fact that pain is not measured to complement anatomical changes with treatment is an issue together with no follow up in the days after therapy which is a clinical reality that provides an insight into clinical benefit.

Reviewer 2 Report

Although it is an article that could be very interesting, it has very important defects that make me reject it.
The first and most important is that in their discussion, one of the aspects on which the authors rely, is on the pressure exerted in the different techniques used, but nevertheless in no case they explain how they objectively measure what pressure they are making with their hands… It cannot be accept a study based on the different pressures without measuring exactly and objectively that pressure.
On the other hand it is a study with only one intervention. Taking into account that this is a journal with a high impact factor, the publication of this article with such non-objective interventions does not seem appropriate to be published.
The introduction basically refers to low back pain, and, although they discreetly mention that there are some studies that document biomechanical changes with the studied techniques, in no case they reveal the current state of the subject. The research objective is not described.
Furthermore, taking into account the sample… the sample should be increased since the previous analysis showed that at least 25 subjects were needed in each group.
The inclusion criteria are neither clear nor referenced, on what exactly is the diagnosis of LBP based?
As for the Osteopathic manipulative treatment, they should explain it more explicitly in order to be replicable.

Reviewer 3 Report

Accept.

Author Response

We have taken into account the notes on the English language and style in our new version of the article. We have proofread and corrected this article. Some unclear phrases have been replaced with more concise language.

Round 2

Reviewer 1 Report

Disregarding the minor issues with experimental design, the authors have modified the manuscript in accordance with comments raised by the reviewers. Grammatical changes have been reviewed and corrected .

Reviewer 2 Report

Although the authors have tried to answer the questions raised, I do not think it is a study likely to be published in an impact journal. If we want to give osteopathy the adjective of science, we must make it more objective, and this study does not reflect any objectivity or replication capacity. The idea of scientific studies is that they can be replicated, and this one does not have this capacity.